# Trust in Science as a Possible Mediator between Different Antecedents and COVID-19 Booster Vaccination Intention: An Integration of Health Belief Model (HBM) and Theory of Planned Behavior (TPB)

**DOI:** 10.3390/vaccines10071099

**Published:** 2022-07-08

**Authors:** Massimiliano Barattucci, Stefano Pagliaro, Chiara Ballone, Manuel Teresi, Carlo Consoli, Alice Garofalo, Andrea De Giorgio, Tiziana Ramaci

**Affiliations:** 1Department of Human and Social Sciences, University of Bergamo, 24129 Bergamo, Italy; 2Department of Neuroscience, Imaging and Clinical Sciences, University of Studies ‘Gabriele d’Annunzio’, 66100 Chieti, Italy; s.pagliaro@unich.it (S.P.); chiara.ballone@unich.it (C.B.); manuel.teresi@unich.it (M.T.); 3Koinè, Interdisciplinary Center for Psychology and Educational Sciences, 00185 Rome, Italy; carloconsoli@icloud.com; 4Faculty of Human and Social Sciences, Kore University of Enna, 94100 Enna, Italy; alice.garofalo@unikorestudent.it (A.G.); tiziana.ramaci@unikore.it (T.R.); 5Faculty of Psychology, eCampus University, 22060 Novedrate, Italy; andrea.degiorgio@uniecampus.it

**Keywords:** vaccination, booster, intention, vulnerability, severity, trust, SARS-CoV-2

## Abstract

As the literature highlights, many health behavior theories try to explain both social and psychological variables influencing an individual’s health behavior. This study integrates insights relative to the antecedents of getting vaccinated from health behavior theories, particularly including the health belief model (HBM), the theory of planned behavior (TPB), and the different socio-demographic factors. Furthermore, we considered the possible mechanism of impact of distrust in science on individuals’ hesitance and resistance to taking up SARS-CoV-2 vaccination in subjects living in Italy. A correlational study of 1095 subjects enrolled when the national vaccination campaign for the third dose was launched. A questionnaire was used to measure: Italian Risk Perception; subjective norm; trust in science, trust in the vaccine; fear of COVID-19; fear of the vaccine; perceived knowledge about SARS-CoV-2; booster vaccination intention. Principal results show that: (i) the positive relationship provided by HBM theory between perceptions of SARS-CoV-2 risk (vulnerability and severity) and intention to have the vaccine, through fear of COVID-19; (ii) the positive relationship between subjective norms and both trust in science and vaccination intention; (iii) that trust in science plays a crucial role in predicting vaccination intention. Finally, the results provided indications about a positive relationship between subjective norms and fear of COVID-19, and a full mediation role of trust in science in the relationships between determinants of both TPA and HBM, fear of COVID-19, and vaccination intention. In conclusion, an individual’s intention (not) to get vaccinated requires the consideration of a plethora of socio-psychological factors. However, overall, trust in science appears to be a key determinant of vaccination intention. Additional strategies promoting healthy behavior are needed.

## 1. Introduction

The importance of vaccination and boosters in a world scenario of an increasing likelihood of pandemics is fundamental both for supranational health policies and for the implementation of public and private communication campaigns and policies [1,2,3]. Vaccination behavior and its antecedent in terms of will (e.g., intention, willingness) are influenced by variables of different nature (e.g., individual, family, social, work, etc.) and this has directed the research of its determinants to an approach mostly by compartment ponds between the different theoretical frameworks [4,5]. Moreover, the intention to get vaccinated is influenced by a plethora of factors: (i) self-protection factors (on one hand, being vaccinated is an action to protect one’s health in the long term, but on the other, it is simultaneously seen as a possible health hazard) [6,7,8]; (ii) socio-relational factors [9]; (iii) political and ideological factors [10,11].

Much of the governmental storytelling around intention—or not—to get vaccinated against SARS-CoV-2 focused on the theme of freedom and individual rights: the act of vaccinating has become in effect a political action, a moment to do one’s duty as citizens in society or to protect the weakest. Conversely, the act of not getting vaccinated has also become the expression of a transversal distrust in government health policies, science, and medicine [12,13,14].

The availability of vaccines, booster frequency, and the complex organization of each national system for SARS-CoV-2 vaccination campaigns, have determined the need to continuously monitor the determinants of vaccination intention to reshape communication and nudging campaigns [15,16]. 

The importance of studying socio-psychological variables to understand vaccination intention and inform effective measures has been advocated [17]. A deeper understanding of the underlying psychology of vaccine-resistant and vaccine-hesitant groups can enhance the potential effectiveness of the public health messages targeting these groups [18].

Research in health psychology has generated multiple health behavior theories (HBT) that identify social and psychological variables influencing an individual’s health behavior [19,20,21,22,23]. In the present paper, we aimed to integrate insights relative to the antecedents of getting vaccinated against SARS-CoV-2 from HBTs, particularly including the health belief model (HBM) [24,25], the theory of planned behavior (TPB) [26,27], and the different socio-demographic factors. More specifically, the research intends, on the one hand, to test a theoretical model that integrates subjective norms, risk perceptions and fear of COVID-19, and trust to explain vaccination intention; on the other hand, understanding the peculiarities of the sub-sample with a low intention of vaccination.

## 2. Literature Background

Three main models have been proposed in the literature to explain the intention to get vaccinated: the 5Cs model of vaccines [28], the health belief model [24,25], and that of social norms, based on the theory of planned behavior [27,29]. Each of these models focuses on specific health protection factors, perceptions of utility, collective dimension, and social responsibility as antecedents of intention to get vaccinated. Almost all of the works that used determinants from all three theoretical models reasoned by assuming that the models were independent and that the variables could not relate to each other [23,30,31]; nevertheless, some authors have underlined the need to integrate insights from the three models to provide a more comprehensive explanation for such a complex behavior/behavioral intention as that represented by the intention/hesitancy to take up the SARS-CoV-2 vaccine [19,32,33].

### 2.1. Health Behavior Theories

Basically, the HBM [34] theorized that health-related behavior is determined by two main classes of variables: the value that each individual attaches to a particular goal and their estimate of the likelihood that a given action will achieve that goal. The model describes four dimensions that are likely to influence an individual’s behavior: perceived susceptibility, severity, benefits, and barriers (for a review, see [35]). HBM has been widely used as a predictor of SARS-CoV-2 behavioral vaccination intention [36,37]. According to HBM, people’s pro-health behaviors are significant predictors of intention to have SARS-CoV-2 vaccines and boosters [38,39]. Perceived vulnerability refers to the belief people have that they are at risk of getting infected with SARS-CoV-2 [40]; perceived susceptibility was identified as an important factor influencing the intention to get vaccinated against SARS-CoV-2 [37,41]. Perceived severity refers to the belief that the consequences resulting from getting the disease are serious for the self and others. Individuals that feel threatened or perceive high levels of risk from SARS-CoV-2 will be more likely to express higher levels of SARS-CoV-2 vaccination intention [42]. Perceived high susceptibility of contracting SARS-CoV-2 and perceived high severity of the negative effects of contracting the virus were associated with increased vaccination intention [37].

In addition to risk perceptions, both within the HBM, and in various contributions that have integrated different theoretical models, numerous authors have highlighted the important role of emotional factors resulting from SARS-CoV-2 in the intention to get vaccinated [25,31,43], and in particular on the fear of COVID-19 in the general population and healthcare workers (HCWs) [44,45,46]. Fear of COVID-19 is undoubtedly correlated with the intention to get vaccinated, but its effect seems to be mediated by attitudes of trust/distrust towards government, science, conspiracy, and so on [47].

From the review of the HBM literature, several confirmations emerge on the positive relationship between perceptions of SARS-CoV-2 risk (mostly concerning vulnerability and severity) and fear of COVID-19 [48,49,50,51,52], and on the positive relationship between fear of COVID-19 and vaccination intention [20,47,53]. Following these relationships, we can formulate several hypotheses (see Figure 1): Hp 1a and Hp 1b: a higher risk perception (higher vulnerability and severity) will be positively related to fear of COVID-19; Hp 2a: a higher fear of COVID-19 will be associated with a lower intention to receive the booster;Hp 2b: a higher fear of COVID-19 will be associated with lower Trust in Science.

Among the studies that analyzed the individual level, the relevance of psychological theories of behavior for vaccine acceptance was shown, such as the theory of planned behavior [54]. Shmueli [33] recently incorporated the TPB with the HBM to explain the intention to get SARS-CoV-2 vaccinated among Israeli adults and found the integration of TPB and HBM significantly increased the explained variance in behavioral intentions.

### 2.2. Theory of Planned Behavior

The TPB is an expectancy-value model used to predict and explain human behavior in specific contexts [29]. TPB posits those attitudes, subjective norms, and perceived behavior control as key antecedents of behavioral intention, which is a proximate predictor of behavior [55,56]. Its utility has been demonstrated in predicting various health-related behaviors including intentions to have genetic testing [57], flu vaccine [33], H1N1 vaccine amongst students [58], HPV vaccine amongst students [59], a future HIV [60] and SARS-CoV-2 vaccination [61,62,63], and to parental decision-making for children’s vaccination [64,65], and actual vaccine uptake [55,61,63,66]. Subjective norms represent the individual’s perceived social pressure to enact that behavior. People are heavily influenced by their perceptions of the beliefs and attitudes of relevant others (e.g., friends and family members) [67]. 

Subjective norms refer to a person’s beliefs about what relevant others think about them engaging in a particular behavior, and whether they would approve of it or not. Subjective norms refer to the extent to which people’s willingness to have the SARS-CoV-2 vaccine is influenced by whether their significant social others approve of them having the vaccine or not. TPB research shows that both perceived descriptive and injunctive norms are positively related to vaccine uptake [63,68,69,70,71]. Coronavirus SARS-CoV-2 vaccination intentions correlate positively with a positive attitude toward vaccination, with perceived subjective norms in favor of vaccination among friends, family [69,72], doctors [73,74], and with high perceived behavioral control [11,27,63,68].

Subjective norms play a role in the relationship between conspiratorial beliefs and vaccination. This is consequential, as research suggests that distrust in science—often underpinned by conspiracy beliefs [75] and specific ideologies [76]—affects health-related attitudes and behaviors. Along similar lines, individuals who distrust science might be less motivated to follow science-based advice about important behavioral guidelines to curb the pandemic [77,78]. 

From the review of TPB of vaccine intention, there is basically a certain uniformity in underlining how subjective norms, and trust in science, play a crucial role in predicting vaccination willingness and other prescribed norms and behaviors [11,67,68,79,80]. As a consequence of this, we formulated the following hypotheses:Hp 3a: Subjective norms will be positively related to intention to get the booster;Hp 3b: Subjective norms will be positively related to trust in science;Hp 4: Trust in science will be positively related to intention to get the booster.

Some research has investigated the relationship between Subjective norms and Fear of COVID 19, but their relationship has not been fully clarified [11,63,81].

Integrating some contributions from the literature [77,82,83,84], it seems more appropriate to consider subjective norms as determining and the fear of COVID-19 as a resulting emotional factor. Considering that, we propose the following further hypothesis:Hp 3c: Subjective norms will be positively related to fear of COVID-19.

### 2.3. 5C Model

Grounded in previous theoretical models, the 5C model aimed to reflect a broad scope of predictors of vaccination intention and booster behavior. The 5C model by Brewer and colleagues [85] includes five possible antecedents of vaccination intention, that is, confidence (trust in science, trust in vaccine effectiveness, safety, and necessity and the system that delivers it), complacency, the calculation, constraints, and collective responsibility [28,86].

Validated studies in vaccine hesitancy [87,88,89,90,91] focus primarily on confidence in vaccines and science/medicine and have stated that it plays a role in explaining vaccination behavior. Confidence “is defined as trust in (i) the effectiveness and safety of vaccines, (ii) the system that delivers them, including the reliability and competence of the health services and health professionals, and (iii) the motivations of policymakers who decide on the need for vaccines” [92] (p. 2). Individuals who lack trust in science and confidence have negative attitudes towards vaccination [1]. 

The trust in science factor has been transversally recognized as one of the factors capable of explaining different behaviors and attitudes of adherence to health behaviors prescribed in a pandemic situation [77,93]. Following literature indications [14,77,79,94,95,96,97], trust in science seems to be a key factor, able to modulate both the emotional and social factors responsible for the intention to get vaccinated. Because of this, we formulate the following hypotheses:Hp 5: Trust mediates the relationship between subjective norms and intention to get vaccinated.Hp 6: Trust mediates the relationship between fear of COVID-19 and intention to get vaccinated.

The proposed theoretical model is presented in Figure 1.

### 2.4. Socio-Demographic Factors

Individuals differed in their intention to get vaccinated according to a number of socio-demographic factors. For instance, males usually reported a higher intention to get vaccinated than females [37,40,98,99,100], although the patterns of such difference vary across national contexts [101].

Moreover, age, education, and income appear to influence the intention to get vaccinated, with older people with higher education and income levels often showing higher vaccination intention [88,102,103,104]. By contrast, unemployed individuals and those with a lower socioeconomic status have often been described as less likely to get vaccinated [105,106,107].

In addition to these factors, there are many other sociodemographic features that can predict vaccine hesitancy and uptake [68]. Unwillingness to have a vaccine is usually found at a higher level among religious minorities, those who are less educated, and residents of rural areas [108]. Regarding health-related predictors, it was found that respondents with chronic conditions at higher risk of SARS-CoV-2 or those who were overweight, as well as those who reported having been vaccinated against flu in the previous season were more likely to accept a SARS-CoV-2 vaccine and booster [33]. Nonetheless, other studies have shown no effect of such socio-demographic factors on the intention to take up SARS-CoV-2 vaccines [33,68,109]. 

## 3. Study Aim

Several authors have stressed the need to integrate insights from the separate models described above to better describe the intention to get vaccinated and, in the case of the present research, to get the SARS-CoV-2 booster vaccine. This integration should be directed to consider the interplay between the different factors that may induce people to accept being vaccinated vs. being reluctant to do so. 

**Figure 1 vaccines-10-01099-f001:**
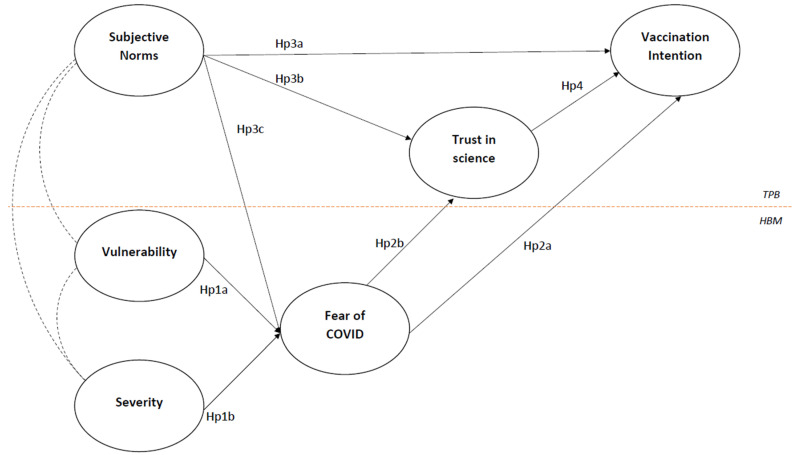
Hypothesized theoretical model.

Based on the literature described above, in the present paper, we aimed to integrate some of the insights provided with regards to vaccination against the SARS-CoV-2 virus in a more comprehensive model in which subjective norms and risk perceptions are considered as antecedents, fear of COVID-19 and trust in science as underlying (emotional) mechanisms, and the intention to have the booster (i.e., the third dose) as the outcome. 

## 4. Materials and Methods

### 4.1. Method

By integrating the Pagliaro [77] and Barattucci [82] models, and following literature indications, the model proposes risk perceptions (vulnerability and severity) and subjective norms as independent variables, fear of COVID-19 and trust in science as mediators, and vaccination intention as an outcome. A correlational study was designed with the aim of exploring the sets of variables associated with booster vaccination intention in an Italian sample, when the national vaccination campaign for the third dose had been launched in Italy. 

Participants completed an online survey between 22 November 2021 (when booster dose administration began, including for people aged between 40 and 59) and 6 December. The overall sampling method was purposeful in that it aimed for diversity in a population that had not already had the third dose, to enhance generalizability. 

Participants were recruited with a short poll (“Will you have the third dose of the SARS-CoV-2 vaccine?”) posted on social media networks (Facebook, LinkedIn, Instagram), direct mailing or through WhatsApp messaging, and by word of mouth, and then provided with a link to a Qualtrics form to fill in a closed-ended questionnaire. Participation was anonymous and voluntary, and there was no way of determining where the participants originated from; the Qualtrics form did track the IPs of participants only to ensure that the same participant did not respond more than once.

Before completing the questionnaire, participants read and accepted a consent form, instructions for participation, and a declaration on data processing in compliance with the Helsinki Declaration (WMA, 2013) and current Italian laws (GDPR). 

The study is part of research that has been approved by the local academic ethics committee (Research project “Return to work and fragile workers”, RN: 03/2020, 28/12/2020). Cases without any data in the scales at use in the present study were considered invalid and the sample size was calculated as the total number of valid returned questionnaires. 

### 4.2. Participants

One thousand and twenty participants began filling in the questionnaire; the final sample was composed of 1095 individuals who adequately completed it. Participants were mainly female (N = 864, 78.9%), mostly with a university or post-graduate degree (academic degree, N = 448, 40.9%; post-graduate, N = 240, 21.9%), currently in work (unemployed, N = 121, 11.1%; student, N = 130, 11.9%), and with a mean age of 40.06 years (SD = 13.8). From a health point of view, participants were mainly vaccinated (N = 1014, 92.6%); among the vaccinated, over 70% had had two vaccines (N = 788, 72%), largely the BioNTech, Pfizer vaccine (N = 609, 55.6%). 

### 4.3. Measures

Participants completed the first section of the questionnaire with socio-demographic information and details about their own SARS-CoV-2 vaccine experiences; then, they filled out a questionnaire made up of the following measures: 

Italian Risk Perception Questionnaire [110] adapted from the original development by Savadori et al. [111]. The researchers adapted the tool to measure 2 single-item dimensions of SARS-CoV-2 infection risk, on a seven-point scale ranging from 0 to 6: severity (perceived seriousness or harmfulness of SARS-CoV-2) and vulnerability (perception of the risk of being infected with SARS-CoV-2); item examples: “Considering the scale (from 1 “not exposed” to 7 “totally exposed”), in your opinion, to what extent do you think you are exposed to SARS-CoV-2 risk?” [112].

Subjective norms were measured with a single-item scale from the literature [113], on a seven-point scale ranging from 1 (totally disagree) to 7 (totally agree) (item example: “People whose opinion is important to me (friends, family, colleagues, etc.) think that I need to have the SARS-CoV-2 vaccine”).

Trust in science [77] was measured with an ad hoc 4-items scale, on a seven-point scale ranging from 1 (totally disagree) to 7 (totally agree) (item example: “We should trust the competence of scientists”). Cronbach’s Alpha = 0.84.

Trust in the vaccine [114,115] was measured with an ad hoc 6-items scale, on a seven-point scale ranging from 1 (totally disagree) to 7 (totally agree) (item example: “I believe the SARS-CoV-2 vaccine is important to reduce or eliminate SARS-CoV-2”). Cronbach’s Alpha = 0.86.

Fear of COVID-19 was measured with an ad hoc 2-item scale from the literature (fear of being infected and fear of spreading the virus to others) [82,116], on a seven-point scale ranging from 1 (not worried at all) to 7 (very worried) (item example: “How worried are you about the chance of being infected with the SARS-CoV-2 virus?”). Cronbach’s Alpha = 0.76.

Fear of the vaccine was measured with an ad hoc single-item scale from the literature [82,117,118] on a seven-point scale ranging from 1 (not worried at all) to 7 (very worried) (item example: “How concerned are you about SARS-CoV-2 vaccination?”).

Perceived knowledge about SARS-CoV-2 was measured with an ad hoc 3-items scale from the literature [2,117,118] on a seven-point scale ranging from 1 (no knowledge) to 7 (excellent knowledge) (item example: “How much do you think you know about the progress of the pandemic in recent weeks?”). Cronbach’s Alpha = 0.67.

Booster vaccination intention was measured with an ad hoc single-item on a seven-point scale ranging from 1 (absolutely not) to 7 (I definitely will), which asked participants: “Are you willing to have the third vaccine, even if you do not fall into those categories required to do so?”. 

Common method variance and method biases were limited with procedures suggested in the literature [119]: the different scales were randomly inserted into the questionnaire and graphically separated from each other, and various scale formats and endpoints were utilized for each of the measured variables. Harman’s single factor test indicated that the first attribute contributes only 30% of the inconsistency in the data [120].

Gender, age, and education were included as control variables. Table 1 presents the descriptive statistics and zero-order correlations among the measured variables; through a Kolmogorov–Smirnov test for each of the variables involved, the distribution of all asymmetry and kurtosis values of the measured variables resulted between −1 and +1.

### 4.4. Data Analysis

Data analysis was performed in three steps. In step 1, a series of individual control variables (gender, age, education, work status, vaccination status, etc.) were tested as predictors of intention to vaccine through multiple regression analysis. In step 2, some specific analyses (see next paragraph) were conducted for the description of the sub-sample of zero intention to get vaccinated. In step 3, the research tested measured variables as predictors of vaccination intention through multiple regression analysis. In step 3, the relationships between measured variables expressed by the proposed theoretical model and the main hypotheses were tested through a comprehensive structural equation model with AMOS22. Commonly reported fit statistics were: comparative fit index (CFI), Tucker–Lewis index (TLI), normed fit index (NFI), goodness of fit index (GFI), incremental fit index (IFI), root mean square error of approximation (RMSEA), plus standardized root mean square residual (SRM) for measurement model fit.

## 5. Results

First of all, we proceeded to evaluate the effect of socio-demographic variables on the intention to vaccine through *t*-tests for independent samples, ANOVA, and multiple regressions. Neither gender nor age differences were found, while the educational qualification was a predictor of the intention to vaccine (*t* = 2.33, *p* < 0.02, R2 = 0.005), as well as the number of vaccines made (*t* = 7.97, *p* < 0.001, R2 = 0.06): as the number of vaccines already made and the qualification increases, a greater intention to vaccinate for the third dose is associated.

### 5.1. Predictors and Clusters for Vaccination Intention

The unvaccinated reported that they choose to be not vaccinated due: (a) to the vaccine risk for health (“The vaccine could have dangerous side effects”; N = 22, 27.2%); (b) to personal preferences regarding the management of personal health (“I prefer to develop immunity from infection rather than immunity through vaccination”, N = 14, 17.3%); (c) to the belief that they did not need it since they had already contracted SARS-CoV-2 in the past (N = 14, 17.3%). Among other factors, it is also necessary to consider other reasons for the declared non-vaccination: on one hand, real limitations, health problems, pregnancy, and breastfeeding; on the other, the SARS-CoV-2 vaccine is considered “experimental and unsafe”, non-vaccination is a way to express dissent with government restrictions and pandemic management, the vaccine gives rise to more variants and is not the appropriate method to contain a pandemic, current scientific paradigms are outdated. 

The analytic setup was based on IBM SPSS Modeler, focusing on the target of subjects with a low propensity to have the third dose, and splitting the sample into two: high (5–7) propensity subjects (N = 895 individuals, 81.74%) and low (1) propensity subjects (N = 200 individuals, 18.26%). The education level distributions of the two sub-samples were significantly different (chi-square = 11.929, df = 3, *p* < 0.05): the lower the educational level, the lower the propensity to have the third dose. When dealing with a low propensity to have the third dose, trust in science is one of the most important factors (see Figure 2): more than 60% of the individuals having low propensity to have the third dose, had low trust in science (<3). Conversely, subjects showing a higher propensity to have the third vaccination (in Figure 2; target high propensity) also declared a higher level of trust in science (5 or more). High propensity to vaccination subjects had an average trust in science of 6.01 (SD = 1.16), while low propensity to vaccination individuals showed an average trust in science of 4.46 (SD = 1.67).

The figure above shows the incidence of high propensity vs. low propensity to vaccination individuals, with respect to trust in science levels (from 0 to 7, in 20 equal-spaced bins). The graph is normalized to 1 in each of the 20 bands in order to better depict the incidence of high propensity vs. low propensity individuals. Low levels of trust in science (≤4) clearly show a significant proportion of low propensity individuals (≥50%) and vice-versa for high propensity individuals with high levels of trust in science. 

### 5.2. Measurement Model

Table 1 shows the descriptive statistics for all the measures and zero-order correlation between them. In order to explore the construct validity and the measurement model, a confirmatory factor analysis (CFA) through SEM was conducted: a comparison of four nested models from one factor to a final model composed of the nine principal latent factors (risk perceptions, trust in science/the vaccine, fear of COVID-19/the vaccine, subjective norms, SARS-CoV-2 knowledge, vaccination intention) is presented in Table 2: considering that five dimensions were measured with single items, there was an evident amelioration of all indices from the first to the final model.

### 5.3. Tested Model

The proposed structural model was tested through SEM (Figure 3): the two risk perception dimensions (vulnerability and severity) and subjective norms as correlated antecedents, with direct relationships with fear of COVID-19 as the intermediate variable, which has direct links with trust in science (as a mediator) and vaccination intention (as an outcome). Consistently with our hypothesized relationships the model showed excellent goodness of fit: chi-square = 10.59 (df = 4; *p* < 0.001), RMSEA = 0.039, CFI = 0.995, IFI = 0.995, NFI = 0.992, GFI = 0.996, TLI = 0.974, with all significant relationships (*p* < 0.001). 

Regression weights are presented in Table 3, while the path diagram of the final model is shown in Figure 3. All of the associations were highly significant (*p* < 0.001). 

As hypothesized (Hp 1a and Hp 1b), both dimensions of risk perception are positively related to fear of COVID-19, as well as for subjective norms (Hp 3c); taking into account direct and indirect effects of antecedents and intermediate variables on vaccination intention, results seem to clearly confirm that the expected effect of subjective norms, and risk perceptions through fear of COVID-19, is fully mediated by trust in science (Table 3; Hp 5 and Hp 6). Overall, the relationships expressed in the model explained 33% of the variance for fear of COVID-19, 23% for trust in science, and 36% for vaccination intention.

## 6. Discussion

Different theoretical frameworks focusing on antecedents of a different nature (social responsibility, beliefs, health protection factors, emotions, etc.) have been considered in the literature to explore the intention to get vaccinated, in particular in conjunction with the global SARS-CoV-2 pandemic scenario [121,122,123]. However, even the studies that considered determinants from the different models assumed that factors could not relate to each other (for example, social influencing factors and affect), highlighting the need for models that can provide a more inclusive explanation for behavioral vaccination intention, integrating cognitions and affect [19,23,32]. 

In the present paper, we tried to integrate insights from different theoretical frameworks to provide a better understanding of the psychological underpinnings of the intention to get vaccinated, with particular reference to the third booster dose. In particular, we proposed and tested a theoretical model where subjective norms have been integrated with risk perceptions and fear of COVID-19 to explain vaccination intention, through the mediation of trust in science. 

In line with the abundant literature on health behaviors, our findings confirmed: (i) the positive relationship provided by HBM theory between perceptions of SARS-CoV-2 risk (vulnerability and severity) and vaccination intention, through fear of COVID-19 [20,47,48,49,50,51,52,53]; (ii) the positive relationship between subjective norms and both trust in science and vaccination intention [124,125]; (iii) that trust in science plays a crucial role in predicting vaccination intention (11,67,68,79,80,83,84]. The results also provided indications about a positive relationship between subjective norms and fear of COVID-19, and a possible full mediation role of trust in science in the relationships between determinants of both TPA and HBM, fear of COVID-19, and vaccination intention.

Some important results also emerge from the analysis of the intention to get vaccinated groups: first, those who did not get vaccinated claim the reason is that the vaccine is not considered safe, mostly due to personal beliefs regarding the danger of the vaccine, the importance of making choices independently, and to be able to freely express their dissent towards the government and health measures in response to the pandemic. Therefore, beyond the possible reasons of personal health (pregnancy, pathologies, past SARS-CoV-2 disease, etc.), people who are more resistant to SARS-CoV-2 vaccination and unvaccinated mostly have negative personal political thoughts that express dissent towards government actions and health policies. Finally, it is necessary to highlight, in line with other evidence in Italy, among the socio-demographic variables, the effect on the intention to vaccine of the educational qualification and the number of vaccines already made [95]. Compared to previous research in the same context, the results of this study, on the one hand, clearly confirm that the Italian is a population, compared to others, which has shown great compliance with vaccination and a high vaccination intention [126,127]. 

On the other hand, our results did not provide confirmation to other evidence that showed the effect of age and gender on the intention to vaccinate [128], while they are consistent with those that highlighted the effect of educational qualification, fear of COVID-19, and trust in science and governments [77,95,127]. 

### 6.1. Limitations and Future Directions

The study reported here has some limitations that should be acknowledged and overcome in future ad hoc studies. First and foremost, as the great bulk of research was conducted with regards to an individual’s reactions to the pandemic, the nature of this study is cross-sectional. Despite previous research demonstrating the causal relationship between the antecedents of the health behaviors that we have considered here and the behavioral intention/actual behavior, we should be cautious when interpreting the associations between variables that we have discussed here. Future ad hoc studies should be designed to ascertain the causal relationship between these variables, for instance by means of longitudinal designs.

In considering the results of this study, it is necessary to take into account the limitation due to the composition of the sample which does not seem to fully represent the population, due to a lack of homogeneity in the qualification and gender. It is not possible to exclude that the results on the effect of trust in science may have been somehow affected by the educational level of the sample collected.

Furthermore, it is necessary to report the possible presence of selection bias due to the fact that the sample has mostly vaccinated people.

A further avenue for future research is the extension of the model we proposed in this paper. Indeed, as we have stressed above, dealing with the pandemic requires the integration of a previously proposed model to provide a better understanding of an individual’s behavior. This means that further integrations are possible and desirable, and that we consider the proposed model as a starting point for a more comprehensive understanding of intention to get vaccinated, rather than the only possible one. In particular, the relationship between variables of the HBM and those of the TPB, and the possible interaction of emotional factors that result from the different cognitive and social assessment processes (SARS-CoV-2 risk, risk of vaccine damage, risk of social and work exclusion, trust in government, trust in vaccines, etc.) will need to be investigated more thoroughly.

### 6.2. Theoretical Implications

The current research contributes to the literature on the factors that most emerge as predictors of vaccination intention, and the possible mechanism of impact of distrust in science on individuals’ hesitance and resistance to the uptake of SARS-CoV-2 vaccination [128,129] in the Italian context. From a theoretical point of view, our results seem to confirm that the expected effect of subjective norms, and risk perceptions through fear of COVID-19, is mediated by trust in science. Following literature indications [14,77,79,94,95,96,97], our results also show the role of trust in science as a key factor, able to modulate both the emotional and social factors responsible for the intention to get vaccinated. Indeed, trust in science consistently emerged as a key determinant of adherence to government indications to avoid the spread of SARS-CoV-2 around the world, as several cross-national studies revealed: this has been found to be the case even regarding the most controversial issue, that is, whether to get vaccinated. As reported above, vaccine-hesitant people refer to distrust in science and scientists to explain their own choices, and this has also been related to conspiracy ideologies in other studies. Therefore, overall, the key role in (dis-)trust in understanding SARS-CoV-2 -related behaviors has been fully confirmed in our study: beyond the social, cognitive, and emotional determinants (fear), it is precisely the trust in science that acts as a driver of vaccination intention. This result also appears consistent in the light of those relating to socio-demographic variables: confidence in science and the intention to get vaccinated for the third dose is related both to the qualification and to the number of vaccine doses already made [95,126].

### 6.3. Practical Implications

Emergency situation conditions and various environmental factors such as critical events (lockdowns) have a major impact on peoples’ perceptions and behaviors [130,131], and since the beginning of the SARS-CoV-2 pandemic, there has been much misinformation and many conspiracy theories that may have increased an individuals’ hesitance and resistance to taking up SARS-CoV-2 vaccination [67,128]. 

The expectations of family, friends, and colleagues, knowledge of the risks of SARS-CoV-2, and the consequent fear of COVID-19, are modulated by the level of trust in science: it therefore seems clear that resistance to vaccination is a type of “political response” to the restrictions and obligations imposed on personal freedoms. 

These results are consistent with those of the research which has shown overall that distrust in science, government, and the healthcare system [129,132,133], misinformation [68], belief in conspiracies [67,134], and conspiracy mentality [9,135,136] contribute significantly to a negative attitude to the vaccine. The lack of trust in science therefore appears to be an aspect that is linked to a wider mistrust and critical attitude of governments, politics, health systems, and corporations, which is often also associated with an affinity to conspiracy theories and the anti-scientific. From the point of view of institutional communication, in the future, it will be necessary firstly to separate health communication from governmental communication, and highlight the advantages in terms of health more than those of adherence to government restrictions (experienced negatively); thus, since a low propensity for having a third dose vaccination seems clearly associated with lower educational levels, particular attention should be given to targeting institutional communication to this section of the population.

## 7. Conclusions

Understanding an individual’s intention (not) to get vaccinated requires the consideration of a plethora of socio-psychological factors. In the present paper, we provided evidence that, among other factors, subjective norms, risk perceptions, fear of COVID-19, and, above all, trust in science and scientists appear to be key determinants of vaccination intention regarding the intention to have the third booster dose among a sample of Italians. The biggest health crisis in recent decades has required health and government interventions for the prevention and containment of the virus that has changed the social life, perceptions, and emotional responses of the population; therefore, understanding the psychological factors that guide and motivate the intentions to get vaccinated appears increasingly crucial and useful at the application level. In addition to the weight of the influence of the social reference groups and of the perceptions of risk as antecedents of the emotional responses that can influence the intention to get vaccinated, the feeling of trust in science (and in institutions, governments, etc.) seems to be central because it appears capable of making people go beyond the simple perception of affective factors.

## Figures and Tables

**Figure 2 vaccines-10-01099-f002:**
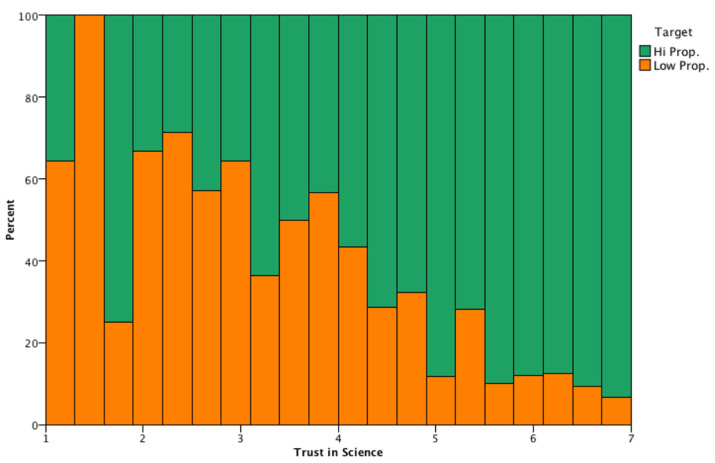
Incidence of high propensity vs. low propensity to vaccination individuals, with respect to trust in science levels.

**Figure 3 vaccines-10-01099-f003:**
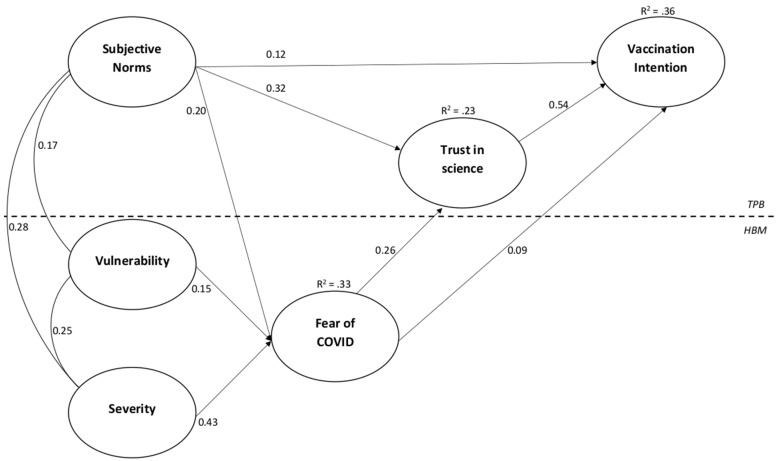
Path diagram of the tested model.

**Table 1 vaccines-10-01099-t001:** Descriptive statistics and zero-order correlations among the variables of the study.

	M (SD)	1	2	3	4	5	6	7	8	9
1. Vulnerability	4.44 (1.5)	-								
2. Severity	4.65 (1.4)	0.250 ***	-							
3. Fear of COVID-19	5.52 (1.4)	0.290 ***	0.522 ***	-						
4. Fear of vaccine	3.06 (1.9)	–0.059	–0.121 **	–0.183 **	-					
5. COVID-19 knowledge	4.70 (1.3)	0.059	0.093 **	0.096 **	0.039	-				
6. Subjective norm	5.58 (1.7)	0.170 **	0.276 ***	–0.340 ***	–0.268 ***	0.041	-			
7. Trust in science	5.72 (1.4)	0.137 **	0.275 ***	0.369 ***	–0.585 ***	–0.006	0.407 ***	-		
8. Trust in vaccine	5.52 (1.4)	0.116 **	0.331 ***	0.368 ***	–0.719 ***	–0.017	0.389 ***	0.763 ***	-	
9. Intention to vaccine	5.86 (1.9)	0.119 **	0.206 **	0.271 ***	–0.482***	0.092 **	0.322 ***	0.585 ***	0.638 ***	-

*** *p* < 0.001; ** *p* < 0.01.

**Table 2 vaccines-10-01099-t002:** Goodness of fit indices of the alternative measurement models on measured variables.

	Chi-Square	df	RMSEA	CFI	IFI	SRMR
Model 1: one factor	3510.733	170	0.134	0.751	0.742	0.110
Model 2: three factors	3464.783	167	0.123	0.756	0.793	0.096
Model 3: six factors	3155.012	164	0.109	0.830	0.809	0.089
Model 4: nine factors	2250.771	161	0.083	0.921	0.909	0.078

df = degrees of freedom; RMSEA = root mean square error of approximation; CFI = comparative fit index; IFI = incremental fit index; SRMR = standardized root mean square residual.

**Table 3 vaccines-10-01099-t003:** Direct and indirect standardized path coefficients (regression weights) of the model.

Direct Effects	
Vulnerability → fear of COVID-19	0.149
Severity → fear of COVID-19	0.430
Subjective norm → fear of COVID-19	0.197
Subjective norm → trust	0.319
Subjective norm → vaccination intention	0.120
Fear of COVID-19 → trust	0.260
Fear of COVID-19 → vaccination intention	0.086
Trust → vaccination intention	0.541
**Indirect Effects via Trust**	
Fear of COVID-19 → vaccination intention	0.140
Subjective norm → vaccination intention	0.217

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
