# Peer review of "Trust in Science as a Possible Mediator between Different Antecedents and COVID-19 Booster Vaccination Intention: An Integration of Health Belief Model (HBM) and Theory of Planned Behavior (TPB)"

_vaccines, 2022, doi:10.3390/vaccines10071099_

Round 1

Reviewer 1 Report

Thank you for the opportunity to review the manuscript titled “Trust in science as a possible mediator between different antecedents and booster vaccination intention: a possible integration of HBM and TPB models”. This is a very interesting topic about  COVID-19 booster vaccination intention however the authors are not clear since the title, due it not mention COVID-19, it seems the paper is about vaccine in general. Moreover the author doesn’t describe Covid-19 in the introduction. I would suggest the author to use a more concise title.

My most important comment, according to my opinion, is the study has one main limitation that the author did not address, this is the socio-demographic features, most of the participant (78,9%) are females with an academic degree (62,8%), the participants may not represent the general population. Moreover, the reading of the literature background is long in comparison to the results that are too short.

I also recommend to review the bibliography, for instance, in the line 91, the refer to article 16, however the 16 reference says “First, perceived susceptibility to COVID-19 vaccine side effects was negatively associated with vaccination intention whereas perceived severity did not show any significant impact. Second, vaccine-related knowledge was not directly related to vaccination intention, but it had an indirect and positive effect on vaccination intention via decreasing perceived susceptibility”. This is not what the authors are claiming in that paragraph.

The authors showed a 92,6% of vaccinated people, is the percentage of unvaccinated participants associated with sociodemographic characteristics? Please clarify. I also wondering if have the authors compared that result with others studies in Italy population. I miss these studies in the discussion.

In the line 89, the authors said “individuals that perceive high levels of risk susceptibility will tend to report lower levels of COVID-19 vaccination intention ”, do the author refers to susceptibility of the disease or of the side effect of the vaccine? Please clarify.

On the line 245 the number is written with letter, for a better reading I recommend numbers, the same that the authors did in the next lines

Author Response

1.      REV s comments

Changes made

This is a very interesting topic about COVID-19 booster vaccination intention however the authors are not clear since the title, due it not mention COVID-19, it seems the paper is about vaccine in general. Moreover, the author doesn’t describe Covid-19 in the introduction. I would suggest the author to use a more concise title.

We really thank you for your words and sorry for being not clear. We agree that the title and the introduction were adding some ambiguity regarding COVID-19.

In this vein and following your indication, we added in the title “COVID-19” and deleted the repetition of the word “possible”.

“Trust in science as a possible mediator between different ante-cedents and COVID-19 booster vaccination intention: an integration of HBM and TPB models”.

Moreover, we added 3 specifications with COVID-19 in the introduction. Really thank you for this suggestion.

My most important comment, according to my opinion, is the study has one main limitation that the author did not address, this is the socio-demographic features, most of the participant (78,9%) are females with an academic degree (62,8%), the participants may not represent the general population.

Dear reviewer, we agree that your comment is totally right. Following your suggestions, we added in the limitation section the following phrase regarding the socio-demographic features:

In considering the results of this study, it is necessary to take into account the limitation due to the composition of the sample which does not seem to fully represent the population, due to a lack of homogeneity in the qualification and gender. It is not possible to exclude that the results on the effect of trust in science may have been some-how affected by the educational level of the sample collected.

Furthermore, it is necessary to report the possible presence of selection bias due to the fact that the sample has mostly vaccinated people. 

Moreover, the reading of the literature background is long in comparison to the results that are too short.

Dear reviewer, we thank you for the suggestion. Anyway, the two sections (results and introduction) are equivalent (3 pages each); due to the different requests made by the other 2 reviewers [even, one of the other reviewers thinks the background of the literature is too short (!)], and due to the requested changes and additions to the article revisions, we could not consistently modify the results according to your specific request. As you can see from the changes made, however, the proportions of the parts have conformed to the required revisions. I hope you can appreciate this change.

I also recommend to review the bibliography, for instance, in the line 91, the refer to article 16, however the 16 reference says “First, perceived susceptibility to COVID-19 vaccine side effects was negatively associated with vaccination intention whereas perceived severity did not show any significant impact. Second, vaccine-related knowledge was not directly related to vaccination intention, but it had an indirect and positive effect on vaccination intention via decreasing perceived susceptibility”.

This is not what the authors are claiming in that paragraph.

Dear reviewer, both the sentence and the quote help create ambiguity of sorts in the speech. We therefore considered deleting the entire sentence and the ref. thanks for the suggestion.

The authors showed a 92,6% of vaccinated people, is the percentage of unvaccinated participants associated with sociodemographic characteristics? Please clarify.

Dear reviewer, we’re sorry for being not clear.

The effect of socio-demographic characteristic is now described in this phrase:

“First of all, we proceeded to evaluate the effect of socio-demographic variables on the intention to vaccine through t-tests for independent samples, ANOVA, and multiple regressions. Neither gender nor age differences were found, while the educational qualification was a predictor of the intention to vaccine (t = 2.33, p <.02, R2 = .005), as well as the number of vaccines made (t = 7.97, p <.001, R2 = .06): as the number of vac-cines already made and the qualification increases, a greater intention to vaccinate for the third dose is associated.”

I also wondering if have the authors compared that result with others studies in Italy population. I miss these studies in the discussion.

Dear reviewer, we are grateful for your suggestion. That is a really important tip to better our discussion.

We added the following phrase

Finally, it is necessary to highlight, in line with other evidence in Italy, among the socio-demographic variables, the effect on the intention to vaccine of the educational qualification and the number of vaccines already made [95]. Compared to previous research in the same context, the results of this study, on the one hand, clearly confirm that the Italian is a population, compared to others, which has shown great compliance with vaccination and a high vaccination intention [ref. 127, 128].

On the other hand, our results did not provide confirmation to other evidence that showed the effect of age and gender on the intention to vaccinate [ref. 128], while they are consistent with those that highlighted the effect of the educational qualification, fear of covid and trust in science and governments [refs 77,95,128].

In the line 89, the authors said “individuals that perceive high levels of risk susceptibility will tend to report lower levels of COVID-19 vaccination intention”, do the author refers to susceptibility of the disease or of the side effect of the vaccine? Please clarify.

Dear reviewer, both the sentence and the quote help create ambiguity of sorts in the speech. We therefore considered deleting the entire sentence and the ref. thanks for the suggestion.

On the line 245 the number is written with letter, for a better reading I recommend numbers, the same that the authors did in the next lines

We modified accordingly to your nice suggestion

Reviewer 2 Report

Thank you for the opportunity to review this interesting article. After reading it I found the following aspects related to:

Abstract. The authors clearly present the research objective, the results obtained and the conclusions of their study.

Introduction. This section is brief but the authors do not clearly state the purpose of their research. I suggest that the authors do this.

Literature background. This section is well organized on the subsections dedicated to the three models considered by the authors when explaining vaccination. Research hypotheses are launched after each section making understanding much easier for readers and researchers.

Study aim. The hypothetical theoretical model launched by the authors and presented in Figure 1 presents the conceptual starting framework of their research.

Materials and methods. The quantitative research was based on a questionnaire. This section clearly presents the data used for the research and the entire methodology developed during the research.

Data Analysis and Results. The results obtained from the testing of the authors are conclusive based on previous input data and using SPSS for data processing.

Discussion. This section should be extended to explain in more detail what was analyzed. Subsections 6.1. Limitations and future directions if 6.2. Theoretical Implications and 6.3. Practical implications should be transferred to the last section on Conclusions. I suggest that the authors do this.

Conclusions. This section should highlight the main contributions made by the authors through the research conducted. I suggest the authors emphasize their contributions.

Author Response

2 REV s comments

Changes made

Thank you for the opportunity to review this interesting article. After reading it I found the following aspects related to:

Thank you for the support! We’re happy of your impression. We’re really grateful.

Abstract. The authors clearly present the research objective, the results obtained and the conclusions of their study.

Thank you very much for your comment.

Introduction. This section is brief but the authors do not clearly state the purpose of their research. I suggest that the authors do this.

We believe that your suggestion is totally right. In this vein and following your indication, we added the following phrase at the end of the introduction:

More specifically, the research intends, on the one hand, to test a theoretical model that integrates subjective norms, risk perceptions and fear of COVID-19, and trust to explain vaccination intention; on the other hand, understanding the peculiarities of the sub-sample with a low intention of vaccination.

Literature background. This section is well organized on the subsections dedicated to the three models considered by the authors when explaining vaccination. Research hypotheses are launched after each section making understanding much easier for readers and researchers.

Thank you for the support! We’re happy of your impression. We’re really grateful.

Study aim. The hypothetical theoretical model launched by the authors and presented in Figure 1 presents the conceptual starting framework of their research.

Thank you very much for your comment.

Materials and methods. The quantitative research was based on a questionnaire. This section clearly presents the data used for the research and the entire methodology developed during the research.

Thank you for the support! We’re happy of your impression. We’re really grateful.

Data Analysis and Results. The results obtained from the testing of the authors are conclusive based on previous input data and using SPSS for data processing.

Thank you very much for your comment.

Discussion. This section should be extended to explain in more detail what was analyzed. Subsections 6.1. Limitations and future directions if 6.2. Theoretical Implications and 6.3. Practical implications should be transferred to the last section on Conclusions. I suggest that the authors do this.

Dear reviewer, we believe that your suggestion is right, since there was many missing detailed information to explain results. In this vein, and following your suggestions we added the following phrase:

discussion section

Finally, it is necessary to highlight, in line with other evidence in Italy, among the socio-demographic variables, the effect on the intention to vaccine of the educational qualification and the number of vaccines already made [95]. Compared to previous research in the same context, the results of this study, on the one hand, clearly confirm that the Italian is a population, compared to others, which has shown great compliance with vaccination and a high vaccination intention [ref].

On the other hand, our results did not provide confirmation to other evidence that showed the effect of age and gender on the intention to vaccinate [ref], while they are consistent with those that highlighted the effect of the educational qualification, fear of covid and trust in science and governments [77, 95].

Limitation section

In considering the results of this study, it is necessary to take into account the limitation due to the composition of the sample which does not seem to fully represent the population, due to a lack of homogeneity in the qualification and gender. It is not pos-sible to exclude that the results on the effect of trust in science may have been some-how affected by the educational level of the sample collected.

Furthermore, it is necessary to report the possible presence of selection bias due to the fact that the sample has mostly vaccinated people. 

In particular, the relationship between variables of the HBM model and those of the TPB, and the possible interaction of emotional factors that result from the different cognitive and social assessment processes (COVID-19 risk, risk of vaccine damage, risk of social and work exclusion, trust in government, trust in vaccines, etc.) will need to be investigated more thoroughly.

Theoretical implication

This result also appears consistent in the light of those relating to socio-demographic variables: confidence in science and the intention to get vaccinated for the third dose is related both to the qualification and to the number of vaccine doses already made [refs].

Conclusions. This section should highlight the main contributions made by the authors through the research conducted. I suggest the authors emphasize their contributions.

Dear reviewer, we believe that you underlined a right suggestion. We tried to emphasize our contribution in the conclusion section, adding this sentence:

The biggest health crisis in recent decades has required health and government interventions for the prevention and containment of the virus that has changed the social life, perceptions, and emotional responses of the population; therefore, understanding the psychological factors that guide and motivate the intentions to get vaccinated ap-pears increasingly crucial and useful at the application level. In addition to the weight of the influence of the social reference groups and of the perceptions of risk as ante-cedents of the emotional responses that can influence the intention to get vaccinated, the feeling of trust in science (and in institutions, governments, etc.) seems to be central, because it appears capable of making people go beyond the simple perception of affective factors.

Reviewer 3 Report

The article's aim was to apply a “comprehensive model in which subjective norms and risk perceptions are considered as antecedents, fear of COVID-19 and trust in science as underlying mechanisms, and the intention to have the third dose as the outcome”. The topic is an interesting one that may contribute to orienting public health policies regarding vaccination. However, there are a number of shortcomings that must be addressed before publication.

Abstract

Line 21. Why was the study classified as correlational?  It does not match the type of study classification that is commonly applied in epidemiology[1].

The hypothesis could be improved for a better understanding of their meaning:

Line 110 -113  Hp 1a and Hp 1b: Risk perceptions (i.e., vulnerability and severity) will be positively related 110 to fear of COVID-19;

Hp 2a: Fear of COVID-19 will be positively related to intention to receive the booster;

Hp 2b: Fear of COVID-19 will be negatively related to Trust in Science

E.i.:  Hp 1: A higher risk perception (higher vulnerability and severity) will be associated with higher fear of  COVID-19

Hp 2: A Higher fear of COVID-19 will be associated with a lower intention to receive the booster

Hp 2 b:  A Higher fear of COVID-19 will be associated with lower Trust in Science

Additionally, Hp 1 a Hp 1 b, are not clear

Line 118 requires a reference supporting the statement.

Lines 163 – 165 in the 5C model description paragraph only 4 C were listed.

Line 220 is not clear

4.1 Method

What is the number of people who received the questionnaire? What percentage of respondents used direct mail? What percentage of respondents used WhatsApp?

4.2 Participants

To learn more about the sample characteristics, add information regarding those that did not finish filling up the questionnaire compared with those who did finish the questionnaire, (e.i, sex, age, media used to answer the questionnaires, etc.)

Thirty percent of the participants had at least two vaccine doses; therefore, some already had a third dose. How could this affect the analysis of the results? As the outcome variable was third dose vaccination intention.  Are the Cronbach’s alpha values reported on lines 270, 273, and 278 derived from the current study or from the original validation of the scales? Please report both.

To construct SEM  was the distribution of the variables considered? Since the scales applied to evaluate the different constructs have dissimilar scales, were the variables included in the SEM standardized?

Results

Table 1 Please improve clarity, and add labels to the table heading. Maybe using the initials of the variables would help. Consider moving table 1 to the results section of the manuscript. A table with the results of each construct evaluated would be useful.  Additionally, a table with the sociodemographic variables and the evaluated constructs by intention to third dose would be useful.

What percentage of respondents indicated that they did not wish to receive a third vaccination dose? Since the information on the number of vaccines received was available, was there any difference in the intention of receiving the third dose between those with one dose compared with their counterparts with two doses?

Consider moving lines 323 -326 to the discussion section.

Table 2. Is the model goodness of fit statistic appropriate for the dichotomous variable? (High vs low vaccination propensity). Is the goodness of fit index described in the table appropriate for dichotomous outcome variables? (High vaccination propensity versus low vaccination propensity)

It would be interesting to have more information on the effect of age and sex regarding vaccination propensity and the main variables in the SEM.

Participants with high and low propensities to receive a third dose were compared or were they analyzed separately? (lines 328-330)

In relation to the main constructs evaluated in the study (HBM and BP, TS), were there differences between those participants having one dose compared with those having two dosses?

Line 343 Please include the figure number.

Table 3 Please Include the p-values of the standardized path coefficients.

Figure 3. Usually, when using SEM diagrams the latent variables are depicted in ovals and the measured variables as squares. It appears that this is not the case in the diagram presented in Figure 3. Additionally, no direct arrows are presented between vulnerability and severity with trust in science. Were these paths included in the model? If not, this consideration should be addressed in the discussion section of the manuscript.

The authors indicated that “results seem to clearly confirm that the expected effect of subjective norms, and risk perceptions through fear of COVID-19, is fully mediated by trust in science (Table 3; Hp  5 and Hp 6) “.  The data presented in Table 3 do not clearly support the conclusions drawn by the authors. A full assessment of the mediation effect cannot be made from the information provided.

In figure 2 coefficients of determination were provided. This coefficient indicates how much variance can be explained by the variables included in the model. However, this interpretation is appropriate when the dependent variable has a normal distribution. Is it appropriate for dichotomous variables?  (lines 376 – 378).

Discussion

In the result section, there is insufficient information to determine whether full mediation took place. As a result, it is not possible to determine whether the conclusion reached is appropriate.

    An in-depth discussion of the possible presence of selection bias and generalizability of the results should be provided under the limitations of the study. In view of the fact that 90% of the study groups received at least one vaccination dose, those unwilling to receive a third vaccination dose are likely to be underrepresented in the study. This issue should also be addressed. Furthermore, discuss the impact the inclusion of highly educated participants would have on the generalizability of the study results, as well as the differences between this education level and the general population. The study found that trust in science plays a significant role in willingness to receive a third vaccination. Could this be explained by the educational level of the participants? 

General comment

An imbalance exists between the number of pages dedicated to introductions and the number of pages dedicated to discussing results. Please consider making the background literature section more concise. 

[1] Rothman KJ, Greenland S, Lash TL. Modern epidemiology. Philadelphia: Lippincott Williams & Wilkins; 2008.

Author Response

3. REV s comments

Changes made

Abstract

Line 21. Why was the study classified as correlational?  It does not match the type of study classification that is commonly applied in epidemiology.

Dear reviewer,

Thank you very much for suggestion. Unfortunately, we’re psychologist and our is not strictly an epidemiological study;

forgive me, but we do not believe that our study should adopt an epidemiological classification. Since our study regards psychosocial factors, we use a study classification system that relates with our social sciences, that is correlational study.

We’re sure that you’ll understand our choice.  

The hypothesis could be improved for a better understanding of their meaning:

Line 110 -113  Hp 1a and Hp 1b: Risk perceptions (i.e., vulnerability and severity) will be positively related 110 to fear of COVID-19;

Hp 2a: Fear of COVID-19 will be positively related to intention to receive the booster;

Hp 2b: Fear of COVID-19 will be negatively related to Trust in Science

E.i.:  Hp 1: A higher risk perception (higher vulnerability and severity) will be associated with higher fear of COVID-19

Hp 2: A Higher fear of COVID-19 will be associated with a lower intention to receive the booster

Hp 2 b:  A Higher fear of COVID-19 will be associated with lower Trust in Science

Additionally, Hp 1 a Hp 1 b, are not clear

We really thank you for the suggestion. We believe that you’re totally right, and that hypotheses should improve. In this vein, and accordingly with your suggestion, we modified hypotheses as suggested.

Line 118 requires a reference supporting the statement.

Lines 163 – 165 in the 5C model description paragraph only 4 C were listed.

Line 220 is not clear.

Thank you very much, we added a reference, added the missing construct for 5C model, and modified line 220.

Method

What is the number of people who received the questionnaire? What percentage of respondents used direct mail? What percentage of respondents used WhatsApp?

Line 252

Direct mail 7%. Whatsapp 16%. Social media 77%.

Participants

To learn more about the sample characteristics, add information regarding those that did not finish filling up the questionnaire compared with those who did finish the questionnaire, (e.i, sex, age, media used to answer the questionnaires, etc.)

Thirty percent of the participants had at least two vaccine doses; therefore, some already had a third dose. How could this affect the analysis of the results? As the outcome variable was third dose vaccination intention.  

Are the Cronbach’s alpha values reported on lines 270, 273, and 278 derived from the current study or from the original validation of the scales?

325 subjects have not properly completed the questionnaire.

The part of the personal data was at the end of the questionnaire. Therefore, much of this data has been lost. About a quarter of this sub-sample (N = 79) provided authorization but did not fill in any fields in the questionnaire.  

We regret having created some possible ambiguity, but we did not understand at what point we would have expressed this data. in the analysis sample, people who had taken the third dose were excluded.

Yes, they are the values ​​related to this study.

To construct SEM was the distribution of the variables considered? Since the scales applied to evaluate the different constructs have dissimilar scales, were the variables included in the SEM standardized?

Dear reviewer, Table 1 presents the descriptive statistics and zero-order correlations among the measured variables; through a Kolmogorov–Smirnov test for each of the variables in-volved, the distribution all Asymmetry and Kurtosis values of the measured variables resulted between -1 and +1.

Obviously, the variables were included standardized for SEM analysis.

Results

Table 1 Please improve clarity, and add labels to the table heading. Maybe using the initials of the variables would help.

Consider moving table 1 to the results section of the manuscript. A table with the results of each construct evaluated would be useful.

Additionally, a table with the sociodemographic variables and the evaluated constructs by intention to third dose would be useful.

What percentage of respondents indicated that they did not wish to receive a third vaccination dose? Since the information on the number of vaccines received was available, was there any difference in the intention of receiving the third dose between those with one dose compared with their counterparts with two doses?

Consider moving lines 323 -326 to the discussion section.

Table 2. Is the model goodness of fit statistic appropriate for the dichotomous variable? (High vs low vaccination propensity).

Is the goodness of fit index described in the table appropriate for dichotomous outcome variables? (High vaccination propensity versus low vaccination propensity)

It would be interesting to have more information on the effect of age and sex regarding vaccination propensity and the main variables in the SEM.

Participants with high and low propensities to receive a third dose were compared or were they analyzed separately? (lines 328-330)

In relation to the main constructs evaluated in the study (HBM and BP, TS), were there differences between those participants having one dose compared with those having two dosses?

Line 343 Please include the figure number.

Table 3 Please Include the p-values of the standardized path coefficients.

Figure 3. Usually, when using SEM diagrams the latent variables are depicted in ovals and the measured variables as squares. It appears that this is not the case in the diagram presented in Figure 3. THESE ARE ALL LATENT VARS!!!

Additionally, no direct arrows are presented between vulnerability and severity with trust in science. Were these paths included in the model?

If not, this consideration should be addressed in the discussion section of the manuscript.

The authors indicated that “results seem to clearly confirm that the expected effect of subjective norms, and risk perceptions through fear of COVID-19, is fully mediated by trust in science (Table 3; Hp  5 and Hp 6) “.  The data presented in Table 3 do not clearly support the conclusions drawn by the authors. A full assessment of the mediation effect cannot be made from the information provided.

In figure 2 coefficients of determination were provided. This coefficient indicates how much variance can be explained by the variables included in the model. However, this interpretation is appropriate when the dependent variable has a normal distribution. Is it appropriate for dichotomous variables?  (lines 376 – 378).

Dear reviewer, thank you for all these precious suggestions.

We moved table 1 as suggested in the result section.

We added a phrase at the beginning of the result section regarding socio-demographical vars:

First of all, we proceeded to evaluate the effect of socio-demographic variables on the intention to vaccine through t-tests for independent samples, ANOVA, and multiple regressions. Neither gender nor age differences were found, while the educational qualification was a predictor of the intention to vaccine (t = 2.33, p <.02, R2 = .005), as well as the number of vaccines made (t = 7.97, p <.001, R2 = .06): as the number of vac-cines already made and the qualification increases, a greater intention to vaccinate for the third dose is associated.

We regret having created some possible ambiguity, but we did not understand at what point we would have expressed this data. in the analysis sample, people who had taken the third dose were excluded.

Dear reviewer, there is no dichotomous variable between the model measures. the variable intention to vaccine “Booster vaccination intention” was measured with an ad hoc single-item on a seven-point scale ranging from 1 (absolutely not) to 7 (I definitely will)

Dear reviewer, you can find this information at the beginning of the result section.

First of all, we proceeded to evaluate the effect of socio-demographic variables on the intention to vaccine through t-tests for independent samples, ANOVA, and multi-ple regressions. Neither gender nor age differences were found, while the educational qualification was a predictor of the intention to vaccine (t = 2.33, p <.02, R2 = .005), as well as the number of vaccines made (t = 7.97, p <.001, R2 = .06): as the number of vac-cines already made and the qualification increases, a greater intention to vaccinate for the third dose is associated.

Dear reviewer, the procedure is described - lines 303:

The analytic setup was based on IBM SPSS Modeler, focusing on the target of sub-jects with a low propensity to have the third dose, and splitting the sample in two: high (5-7) propensity subjects (N = 895 individuals, 81.74%) and low (1) propensity subjects (N = 200 individuals, 18.26%).

the predictive capacity of the number of vaccine doses in the intention to vaccinate was calculated. Lines 324

Thank you for the suggestion. We added the following phrase:

All of the associations were highly significant (p < 0.001).

Dear reviewer, those depicted are all latent variables.

Ours is an attempt to integrate models and the HBM model has found evidence for the relationship between risk perceptions and fear. the review in the literature did not produce any previous evidence regarding this relationship (from a theoretical point of view); having to use the HPB block we have tested. from the correlations, moreover, it can be deduced that relations are low, and the testing through the sems of the model did not provide significant values.

Dear reviewer, in Table 3 you can read direct and indirect effects.

These are our methodological references. What more would you recommend in the outputs to highlight the mediating effect of trust?

Agler R and De Boeck P (2017) On the Interpretation and Use of Mediation: Multiple Perspectives on Mediation Analysis. Front. Psychol. 8:1984. doi: 10.3389/fpsyg.2017.01984

Byrne, B. M. (2010). Structural equation modeling with amos : basic concepts, applications, and programming (2nd ed., Ser. Multivariate applications series). Routledge.

Dear reviewer, no variable is dichotomous.

 Discussion

An in-depth discussion of the possible presence of selection bias and generalizability of the results should be provided under the limitations of the study.

In view of the fact that 90% of the study groups received at least one vaccination dose, those unwilling to receive a third vaccination dose are likely to be underrepresented in the study. This issue should also be addressed.

Furthermore, discuss the impact the inclusion of highly educated participants would have on the generalizability of the study results, as well as the differences between this education level and the general population.

The study found that trust in science plays a significant role in willingness to receive a third vaccination. Could this be explained by the educational level of the participants? 

Dear reviewer, we believe that your suggestion is right.

We added the following lines to the limitation section

“In considering the results of this study, it is necessary to take into account the limitation due to the composition of the sample which does not seem to fully represent the population, due to a lack of homogeneity in the qualification and gender. It is not possible to exclude that the results on the effect of trust in science may have been some-how affected by the educational level of the sample collected.

Furthermore, it is necessary to report the possible presence of selection bias due to the fact that the sample has mostly vaccinated people.” 

In particular, the relationship between variables of the HBM model and those of the TPB, and the possible interaction of emotional factors that result from the different cognitive and social assessment processes (COVID-19 risk, risk of vaccine damage, risk of social and work exclusion, trust in government, trust in vaccines, etc.) will need to be investigated more thoroughly.

General comment

An imbalance exists between the number of pages dedicated to introductions and the number of pages dedicated to discussing results. Please consider making the background literature section more concise. 

Dear reviewer, we thank you for the suggestion. Anyway, the two sections (discussion and introduction) are equivalent (3 pages each); due to the different requests made by the other 2 reviewers [even, one of the other reviewers thinks the background of the literature is too short (!)], and due to the requested changes and additions to the article revisions, we could not consistently modify the discussion according to your specific request. As you can see from the changes made, however, the proportions of the parts have conformed to the required revisions. I hope you can appreciate this change. Discussion has been really developed.

Round 2

Reviewer 1 Report

Thank you for the clarifications